# Inhibition of DYRK1A disrupts neural lineage specificationin human pluripotent stem cells

Stephanie F Bellmaine[1,2,3], Dmitry A Ovchinnikov[4], David T Manallack[5], Claire E Cuddy[2], Andrew G Elefanty[6,7,8], Edouard G Stanley[6,7,8], Ernst J Wolvetang[4†], Spencer J Williams[1,3†], Martin Pera[2,9†*]

[1]School of Chemistry, University of Melbourne, Victoria, Australia; [2]Department of Anatomy and Neuroscience, University of Melbourne, Victoria, Australia; [3]Bio21 Molecular Science and Biotechnology Institute, University of Melbourne, Victoria, Australia; [4]Australian Institute for Bioengineering and Nanotechnology, University of Queensland, Brisbane, Australia; [5]Monash Institute of Pharmaceutical Sciences, Faculty of Pharmacy and Pharmaceutical Sciences, Monash University, Victoria, Australia; [6]Murdoch Childrens Research Institute, The Royal Children's Hospital, Victoria, Australia; [7]Department of Anatomy and Neuroscience, Faculty of Medicine, Dentistry and Health Sciences, University of Melbourne, Victoria, Australia; [8]Department of Anatomy and Developmental Biology, Faculty of Medicine, Nursing and Health Sciences, Monash University, Victoria, Australia; [9]Walter and Eliza Hall Institute of Medical Research, Florey Neuroscience and Mental Health Institute, Victoria, Australia

*For correspondence:
mpera@unimelb.edu.au

[†]These authors contributed equally to this work

**Abstract** Genetic analysis has revealed that the dual specificity protein kinase DYRK1A has multiple roles in the development of the central nervous system. Increased *DYRK1A* gene dosage, such as occurs in Down syndrome, is known to affect neural progenitor cell differentiation, while haploinsufficiency of *DYRK1A* is associated with severe microcephaly. Using a set of known and newly synthesized DYRK1A inhibitors, along with CRISPR-mediated gene activation and shRNA knockdown of *DYRK1A*, we show here that chemical inhibition or genetic knockdown of *DYRK1A* interferes with neural specification of human pluripotent stem cells, a process equating to the earliest stage of human brain development. Specifically, DYRK1A inhibition insulates the self-renewing subpopulation of human pluripotent stem cells from powerful signals that drive neural induction. Our results suggest a novel mechanism for the disruptive effects of the absence or haploinsufficiency of *DYRK1A* on early mammalian development, and reveal a requirement for *DYRK1A* in the acquisition of competence for differentiation in human pluripotent stem cells.
DOI: https://doi.org/10.7554/eLife.24502.001

## Introduction

The dual-specificity tyrosine-phosphorylation-regulated (Dyrk) kinases belong to a family collectively referred to as CMGC kinases (*Aranda et al., 2011*) that includes cyclin dependent kinases, mitogen associated protein kinases, glycogen synthase kinases, and cyclin dependent-like kinases The distinguishing biochemical features of Dyrk kinases are their ability to phosphorylate serine, threonine and tyrosine residues, and the autoregulation of their kinase activity through tyrosine phosphorylation. The five mammalian Dyrk kinases are divided into two classes. The Class 1 kinase DYRK1A has been implicated in a diverse variety of biological processes, including central nervous system

development, Down syndrome, cancer, beta cell proliferation and diabetes, and Alzheimer's disease, and the discovery of novel DYRK1A inhibitors has been a goal of many recent studies (*Abbassi et al., 2015*; *Aranda et al., 2011*; *Shen et al., 2015*; *Smith et al., 2012*; *Stotani et al., 2016*).

DYRK1A has multiple roles in central nervous system development (*Tejedor and Hämmerle, 2011*). Genetic studies in mice (*Fotaki et al., 2002*) and man (*Bronicki et al., 2015*; *Courcet et al., 2012*; *Dang et al., 2017*; *DDD Study et al., 2017*; *Ji et al., 2015*; *Møller et al., 2008*; *van Bon et al., 2016*; *Yamamoto et al., 2011*) have revealed that haploinsufficiency of *DYRK1A* can lead to severe disorders of brain development, including microcephaly, as well as a generalized developmental delay. *DYRK1A* lies within the Down syndrome critical region on chromosome 21, and an excessive gene dosage of *DYRK1A* is thought to account for some of the central nervous system phenotypes of this disorder (*Duchon and Herault, 2016*). Studies of DYRK1A overexpression have elucidated some of its functions during neurogenesis. In embryonic neuroepithelium, a transient increase in DYRK1A expression results in the cessation of the proliferative divisions that expand the progenitor compartment, and premature entry of these cells into a pro-differentiation neurogenic pathway (*Hämmerle et al., 2011*). In several model systems, DYRK1A overexpression led to exit of neural stem cells from the cell cycle, through mechanisms involving cyclin D1 and p53 (*Najas et al., 2015*; *Park et al., 2010*; *Soppa et al., 2014*; *Yabut et al., 2010*). DYRK1A gene dosage also affects later stages of neurogenesis, including neuronal dendritogenesis (*Benavides-Piccione et al., 2005*; *Göckler et al., 2009*). DYRK1A has also been implicated in tau protein phosphorylation in the pathogenesis of Alzheimer's disease (*Coutadeur et al., 2015*).

Previously we showed that the indole derivative ID-8, in combination with WNT3A, could maintain human embryonic stem cells (hESC) in long-term culture under defined conditions in the absence of exogenous activators of the nodal or FGF signalling pathways, both of which are generally considered to be essential for human pluripotent stem cell (hPSC) maintenance (*Hasegawa et al., 2012*). In the presence of WNT3A, ID-8 modestly enhanced hESC plating efficiency, and strongly inhibited the induction of lineage specific differentiation genes normally observed following WNT treatment of undifferentiated stem cells. Using affinity chromatography, we found that ID-8 bound to Dyrk family members DYRK2 and DYRK4 in extracts of human pluripotent stem cells. We further showed that stable knockdown of *DYRK1A* and *DYRK2* caused a modest increase in the plating efficiency of hESC, but we did not establish whether this effect was related to enhancement of attachment and survival, or to inhibition of differentiation. Thus although these studies suggested an important action of ID-8 on hESC through modulation of Dyrk kinase activity, the actual molecular target of the compound related to its specific biologic activities remained unclear.

In this study we examine the biological activity of ID-8 and a related series of novel indole compounds to determine the role of Dyrk kinase inhibition in stem cell regulation. Human kinome screening, structure activity relationships and targeted gene activation and inactivation studies implicate DYRK1A as the biologically significant target of ID-8. We show that DYRK1A inhibition results in a block to neural specification of human embryonic stem cells. This block is not a uniform response across the entire hPSC population, but instead reflects the ability of DYRK1A inhibitors to insulate the self-renewing subpopulation of hESC from powerful differentiation induction signals. We consider these results in the context of stem cell fate determination, and the deleterious effects of *DYRK1A* loss on central nervous system development.

## Results

### Specificity of a series of indole kinase inhibitors for DYRK1A

We examined the specificity of kinase inhibition by ID-8 (*Figure 1a*) and a related series of novel indole compounds using a biochemical in vitro assay. ID-8 was screened against a panel of 339 human protein kinases by measuring incorporation of radioactive ATP into appropriate substrates (Reaction Biology). Activity (based on percentage inhibition at a 10 µM dose of ID-8) against the top ten protein kinase targets, and several members of the CMGC family, are listed in *Figure 1b*. The specificity of the compound is displayed in a kinome inhibition map in *Figure 1c* (for the complete results of the screen, see *Supplementary file 1*). ID-8 indeed showed selectivity against the CMGC kinase family, with DYRK1B, GSK3B and DYRK1A being the top three kinase targets. Although a

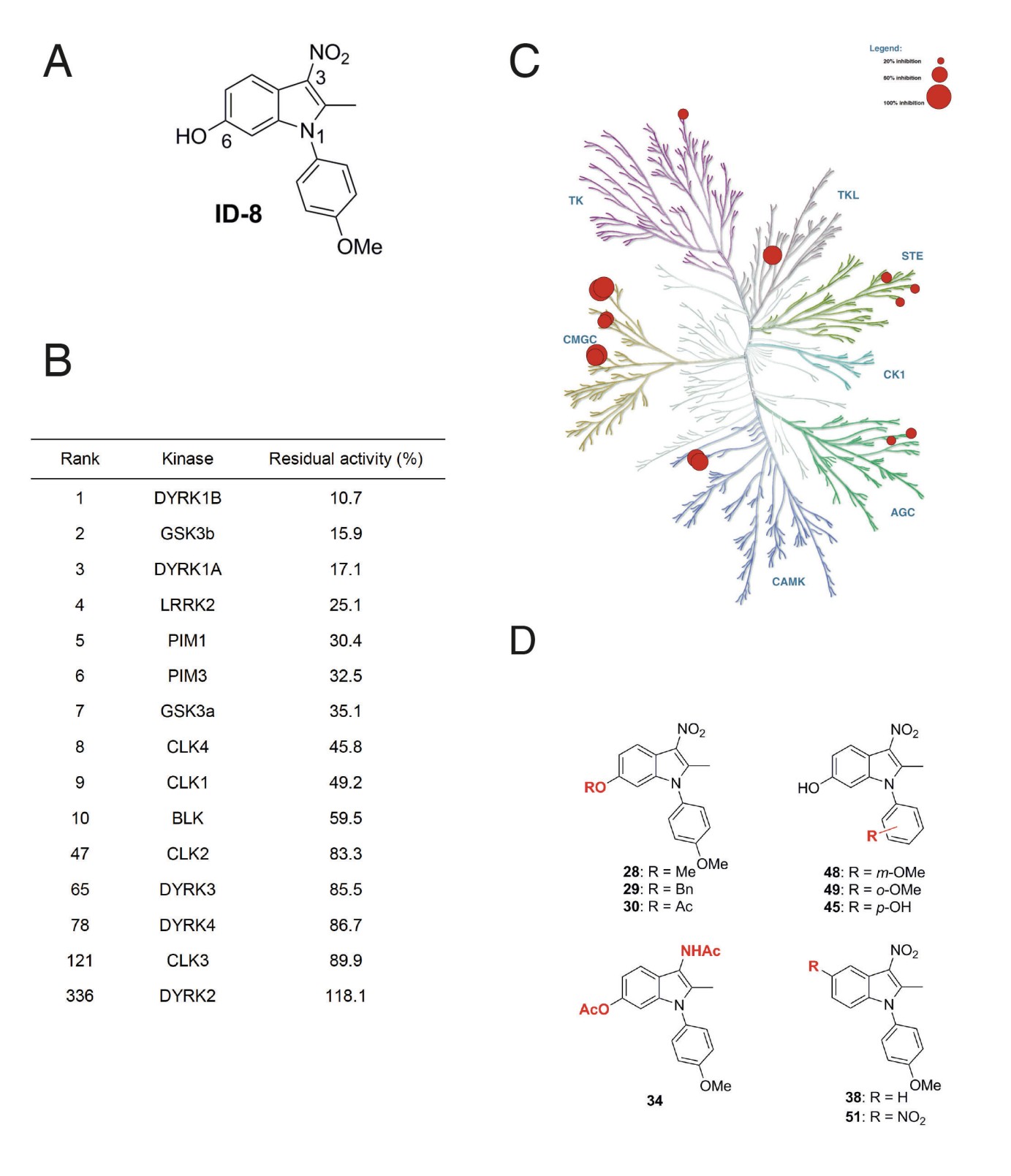

**Figure 1.** Protein kinase inhibition by ID-8. (**A**) ID-8 structure. (**B**) top kinase targets of ID-8 by degree of inhibition at 10 µm, and inhibition of other CMGC group members. (**C**) kinome tree illustrating the extent of protein kinase inhibition in six families of protein kinases. Illustration reproduced courtesy of Cell Signalling Technology, Inc. (www.cellsignal.com). (**D**) Structures of indole compounds related to ID-8.

DOI: https://doi.org/10.7554/eLife.24502.002

*Figure 1 continued on next page*

*Figure 1 continued*

The following figure supplement is available for figure 1:

**Figure supplement 1.** Schematic illustration of chemical syntheses of ID-8 analogues.

DOI: https://doi.org/10.7554/eLife.24502.003

biotinylated derivative of ID-8 bound DYRK2 and DYRK4 in affinity chromatography pull down assays (*Hasegawa et al., 2012*), ID-8 itself showed little activity against these kinases, or against DYRK3. Next we determined $IC_{50}$ values for a subset of these kinase targets, using the same $^{33}P$ incorporation assay (*Table 1*). This study revealed the high specificity of ID-8 for DYRK1A and DYRK1B, and confirmed a lack of activity against DYRK2 and DYRK4. Inhibition of PIM1, GSK3A, GSK3B, and CLK4 at $IC_{50}$ values between 280–450 nM was also observed.

We next synthesised a series of ID-8 derivatives modified at the 1, 3 and 6 positions (*Figure 1d*). The 1 and 6 positions were targeted because modelling studies (discussed below) implicated the –OH and –$OCH_3$ groups as critical to binding at the active site of DYRK1A. The $NO_2$ group was modified since this group might lead to unwanted pharmacological properties in vivo, such as metabolic conversion to toxic nitroso or hydroxylamine derivatives. These compounds were prepared using: (1) the Nenitzescu reaction used for the preparation of ID-8 itself (for compounds **48** and **49**); (2) by derivatization of ID-8 (for compounds **28, 29, 30, 34, 35**); or (3) by Fischer indole synthesis (compounds **38** and **51**). The synthetic pathways used to form these analogues are shown in *Figure 1—figure supplement 1* and described in detail in *Supplementary file 2*.

Biochemical screening of the analogues against the top ten kinase hits for the parent ID-8 are shown in *Table 2*. Compounds **28, 30** and **48** showed excellent selectivity for DYRK1A and DYRK1B. In particular, compound **28** showed much less activity against DYRK1B, CLK1, GSK3B or LRRK2, relative to ID-8. Compound **30** showed better selectivity against DYRK1A relative to CLK1, CLK4 or GSK3A. Compound **48** showed a similar spectrum of kinase inhibition to ID-8 but was less potent against DYRK1A and DYRK1B. Compound **45** showed activity against all of the kinases in the selected panel and maintained high potency of inhibition of DYRK1A and DYRK1B. Considering the modifications responsible for DYRK1A inhibition, these structure-activity results indicate that that inhibition is very sensitive to modifications at the 6-OH group. Installation of nitro group at the 5-position (compound **51**) ablated activity against all kinases tested, except for PIM1 and PIM2. Finally, substitution of the $NO_2$ group at position 3 by an acetylamino group (Compound **34**) eliminated activity against all kinases tested.

**Table 1.** $IC_{50}$ values for ID-8 against a selection of kinases.
Ranks are given based on the original screening data presented in *Figure 1B* and *Supplementary file 1*. NC = an $IC_{50}$ value could not be calculated.

| Rank | Kinase | $IC_{50}$ (nM) |
|---|---|---|
| 1 | DYRK1B | 54 |
| 3 | DYRK1A | 78 |
| 5 | PIM1 | 280 |
| 7 | GSK3a | 380 |
| 8 | CLK4 | 440 |
| 2 | GSK3b | 450 |
| 9 | CLK1 | 4200 |
| 47 | CLK2 | NC |
| 65 | DYRK3 | NC |
| 78 | DYRK4 | NC |
| 121 | CLK3 | NC |

DOI: https://doi.org/10.7554/eLife.24502.004

**Table 2.** Activity of novel indole compounds against DYRK1A and related kinases relative to ID-8.

| Compound | $R_1$ | $R_2$ | $R_3$ | IC$_{50}$ (µM) | | | | | | | | | |
|---|---|---|---|---|---|---|---|---|---|---|---|---|---|
| | | | | DYRK1A | DYRK1B | DYRK2 | CLK1 | CLK4 | GSK3α | GSK3β | LRRK2 | PIM1 | PIM3 |
| ID-8 | 6-OH | NO$_2$ | p-OCH$_3$ | 0.104 | 0.040 | NC | 1.37 | 1.05 | 0.428 | 0.153 | 0.100 | 0.376 | 0.176 |
| 34 | 6-OCOCH$_3$ | NHCOCH$_3$ | p-OCH3 | NC | NC | NC | NC | NC | NC | NC | NC | NC | NC |
| 28 | 6-OCH$_3$ | NO$_2$ | p-OCH$_3$ | 0.346 | 0.695 | NC | NC | 2.23 | 2.36 | 3.44 | 0.854 | 0.176 | 0.259 |
| 29 | 6-OCH$_2$C$_6$H$_5$ | NO$_2$ | p-OCH$_3$ | 6.52 | 3.57 | NC | NC | NC | 18.0 | 14.3 | 8.16 | 18.5 | 8.28 |
| 30 | 6-OCOCH$_3$ | NO$_2$ | p-OCH$_3$ | 0.680 | 0.299 | NC | NC | 11.95 | 22.15 | 1.71 | 0.891 | 4.15 | 1.21 |
| 45 | 6-OH | NO$_2$ | p-OH | 0.122 | 0.060 | 2.30 | 0.364 | 0.163 | 0.375 | 0.233 | 0.046 | 0.243 | 0.180 |
| 48 | 6-OH | NO$_2$ | m-OCH$_3$ | 0.594 | 0.257 | NC | 13.0 | 4.77 | 2.32 | 0.692 | 2.17 | 0.654 | 0.593 |
| 49 | 6-OH | NO$_2$ | o-OCH$_3$ | 0.829 | 0.341 | 41.9 | 10.7 | 8.59 | 6.12 | 3.82 | 1.98 | 1.44 | 0.892 |
| 51 | 5-NO$_2$ | NO$_2$ | p-OCH$_3$ | NC | NC | NC | NC | NC | NC | NC | NC | 7.62 | 4.57 |
| 38 | H | NO$_2$ | p-OCH$_3$ | NC | NC | NC | NC | NC | NC | NC | NC | NC | 20.74 |

DOI: https://doi.org/10.7554/eLife.24502.005

## DYRK1A inhibition blocks neural specification of hESC

In our previous study (*Hasegawa et al., 2012*), we established that ID-8 treatment modestly increased the replating efficiency of hESC. This effect was further enhanced by addition of WNT3A. WNT3A alone promoted propagation of hESC, but also strongly induced expression of differentiation markers. The addition of ID-8 to WNT3A- treated cells partially suppressed the induction of differentiation markers seen in the presence of WNT3A alone. We therefore first investigated the effects of ID-8 alone on hESC cultured under conditions that promote self-renewal. Propagation of hESC in the presence of ID-8 did not affect the cell cycle distribution of hESC (*Figure 2—figure supplement 1a–b*), or the overall distribution of the proportion of cells in the culture bearing stem cell surface markers (*Figure 2—figure supplement 1c*). There was no consistent inhibition of apoptosis by ID-8 (data not shown).

Since we observed previously that the most striking effect of ID-8 on hESC was to inhibit the expression of early lineage specification markers induced by WNT3A, we decided to investigate the effects of the compound on lineage specification in more depth, and to assess the role of DYRK1A inhibition in this effect. Given the importance of DYRK1A in the development of the nervous system, we chose to examine the effects of ID-8 and its homologues on early neural specification in hESC. We used the combined inhibition of BMP and nodal/activin signalling (*Chambers et al., 2009*) to induce neural specification, and assayed the conversion of hESC to neural progenitors by flow cytometry using a PAX6 reporter cell line developed in our laboratories (*Figure 2a*, *Figure 2—figure supplement 2*, and Materials and methods).

Dual SMAD inhibition led to the appearance of neural rosettes in control cultures and the induction of PAX6 as expected (*Figure 2b–c*). However, we observed a clear morphological difference between control cells that had been induced with dual SMAD only versus cells that also received ID-8. The former had formed obvious rosette structures characteristic of early neural differentiation by Day 16, whereas the latter more nearly resembled pluripotent stem cells (*Figure 2b*). Flow cytometry analysis showed that the addition of dual SMAD inhibitors led to the induction of the early human neural progenitor marker PAX6 (*Zhang et al., 2010*) in about 75% of cells after 16 days. By contrast, in the presence of ID-8, there was a dose dependent reduction in PAX 6 induction to around 5% of control levels at 5 µM (*Figure 2c*). We studied the time dependency of the ID-8 effect, to determine whether the compound was required during neural induction or later, after transfer to neural progenitor media. ID-8 had to be present during the period of neural induction and neural progenitor formation to achieve full inhibition of PAX6 expression (*Figure 2d*); it had no effect if added after the neural induction period. The decrease in PAX6 expression caused by ID-8 was accompanied by a decrease in NESTIN staining and an increase in POU5F1 staining in treated colonies (*Figure 2e*). We also evaluated the expression of stem cell surface markers in cultures subjected to neural induction alone or in the presence of ID-8. We have previously shown that flow cytometry using the cell surface markers GCTM-2 and CD9 can identify a continuum of pluripotent stem cell states within the

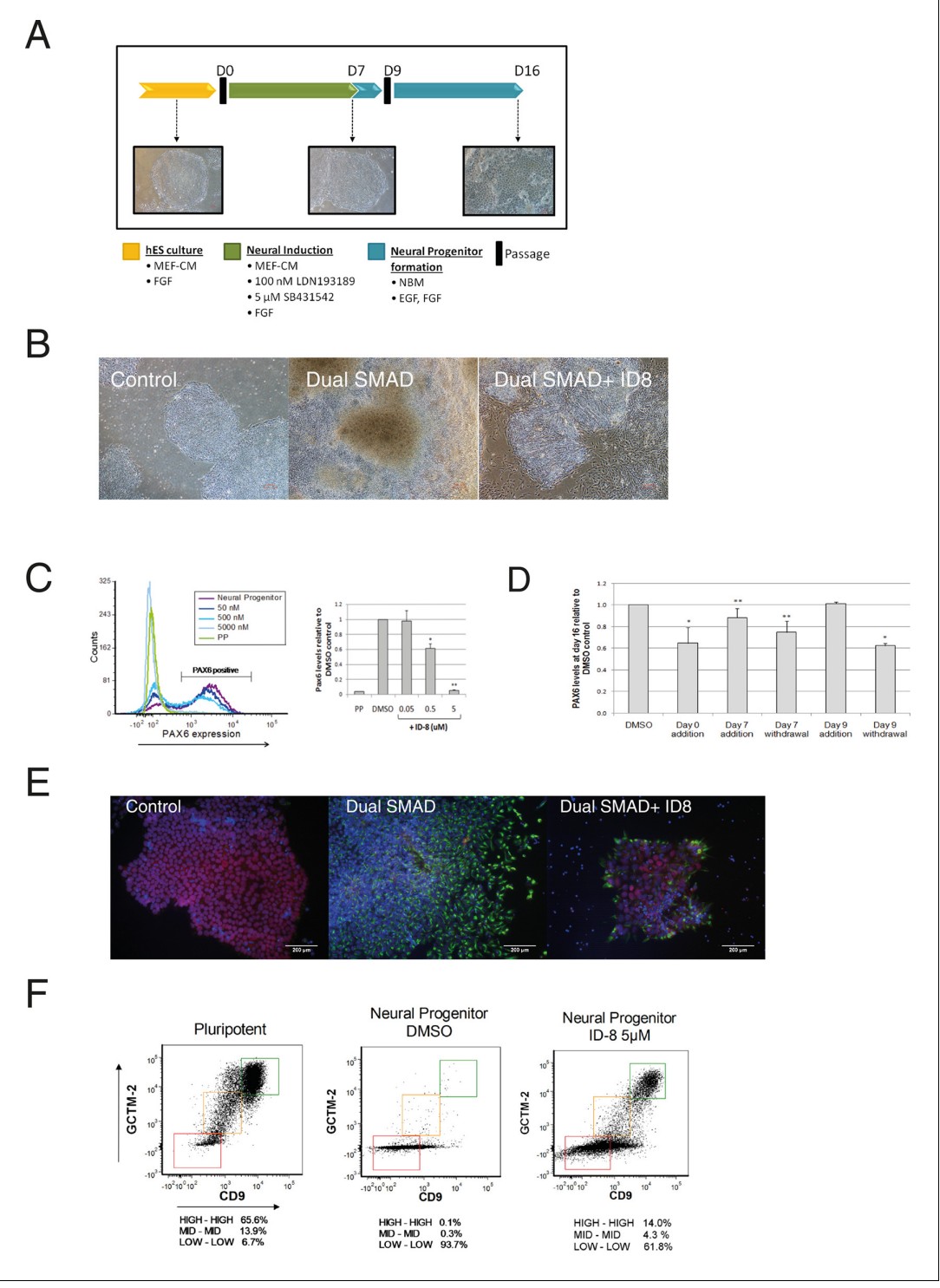

**Figure 2.** Inhibition of neural specification of hESC by ID-8. (**A**) design of time course experiments. Dual SMAD inhibitor induction was carried out from Day 0–7 and cells were assayed for PAX6 expression on Day 16; ID-8 was added from Day 0–16, Day 0–7, Day 0–9, or Day 9–16. (**B**) phase contrast micrographs showing morphology of control hESC, neural progenitors induced by dual SMAD inhibition, and neural progenitors incubated with 0.5 μM ID-8 throughout the neural induction protocol, on Day 16. (**C**) dose response study of the inhibition of induction of PAX 6 positive cells by ID-8 showing flow cytometry profile (left panel) and percentage of PAX6 positive cells (right panel). Error bars, SD; *p<0.05, **p<0.0.01. (**D**) effect of timing of ID-8 exposure at 0.5 μM on inhibition of PAX6 induction. Error bars, SD; **p<0.05, *p<0.0.01. (**E**) indirect immunofluorescence analysis of NESTIN (green) and

*Figure 2 continued on next page*

*Figure 2 continued*

POU5F1 expression (red) in control hESC, neural progenitors, and cultures subjected to neural induction in the presence of 5.0 µM ID-8. Nuclear counterstain, dark blue. (F) flow cytometry profiles showing expression of stem cell surface molecules GCMT-2 and CD9 in control cells, neural progenitors and cultures subjected to neural induction in the presence of 5.0 µM ID-8. A-D, studies carried out with HES3 (PAX6$^{mCherry}$) cell line; E-F, WA09 hESC.

DOI: https://doi.org/10.7554/eLife.24502.006

The following figure supplements are available for figure 2:

**Figure supplement 1.** ID-8 treatment does not interfere with hESC proliferation or alter expression of stem cell markers under conditions that promote self-renewal.

DOI: https://doi.org/10.7554/eLife.24502.007

**Figure supplement 2.** Targeting of the PAX6 gene with an mCherry-Ires-Puro reporter cassette.

DOI: https://doi.org/10.7554/eLife.24502.008

---

population (*Hough et al., 2014*). Induction with dual SMAD inhibition alone resulted in a rapid depletion of the entire pluripotent stem cell population (*Figure 2f*). The addition of ID-8 had an unusual effect. As in control cells, dual SMAD inhibition resulted in loss of stem cell marker expression across much of the population, but the fraction expressing the highest level of stem cell markers persisted (*Figure 2f*). The proportion of cells remaining in this fraction relative to the rest of the population was markedly increased (140 fold) when ID-8 was present during dual SMAD inhibition.

## Effects of known DYRK1A and GSK3β inhibitors and ID-8 analogues on neural specification of hESC

We then examined the effect of a series of known inhibitors of DYRK1A (*Figure 3a*) for effects in the PAX6 induction assay. In the same experiments, we also tested a known inhibitor of GSK3ß, to determine if ID-8 activity against this kinase (noted above) could account for its effects on neural induction. CHIR99021, a highly specific GSK3ß inhibitor, is used in some protocols for maintenance of hESC and therefore might be expected to mimic the inhibition of differentiation observed with ID-8 if the latter were acting through the GSK3B pathway. Of the compounds tested, only harmine and CHIR99021 produced statistically significant inhibition of neural induction in the dose tested (*Figure 3b*). Although some of these previously studied compounds are equipotent inhibitors of DYRK1A compared to ID-8, it is notable that ID-8 and harmine show relative selectivity for DYRK1A and B over DYRK2 and DYRK4 (*Supplementary file 3*).

We next tested our series of novel indole derivatives of ID-8 for their activity in inhibiting PAX-6 induction. Only compounds **28**, **30** and **45** displayed activity at 0.5 µM in this assay (*Figure 3c*). Thus, activity in this assay was quite sensitive to any alteration of the parent compound structure. As observed with ID-8, treatment of cells with compound **28** during dual SMAD inhibition resulted in colonies that contained few NESTIN positive cells and an increase in POU5F1 positive cells (*Figure 3d*). Considering the overall spectrum of kinase inhibition represented in the panel, the results argue that inhibition of DYRK1A accounts for compound activity in the assay. Compounds **34**, **51** and **38** were inactive against DYRK1A and showed no activity in the differentiation assay. Compound **51** retained activity against PIM1 and PIM3, suggesting that off-target effects on these enzymes were unrelated to blockade of neural induction. Compound **28** showed much reduced activity against DYRK1B and GSK3B, and no activity against CLK1, but it was still active in the assay. Compound **30** likewise showed negligible activity against CLK1 and much reduced activity against GSK3A. Compounds **48** and **49**, inactive in the neural induction assay, were active against CLK4 and PIM1 and PIM2.

Although CHIR99021 treatment did reduce the extent of PAX-6 induction by dual SMAD inhibition, unlike ID-8 CHIR99021 co-administration did not result in retention of a pluripotent phenotype. Following neural induction, colonies formed in the presence of CHIR contained fewer POU5F1 positive cells compared to colonies induced in the presence of ID-8 or compound **28** (*Figure 3d*). These results were corroborated by studies of stem cell surface maker expression (*Figure 3e*). Expression of the cell surface antigens identified by antibodies GCTM-2 and TG30 (anti-CD9) was abolished by dual SMAD inhibition. Cells subjected to dual SMAD inhibition in the presence of ID-8 or compound **28** retained stem cell surface markers, but cells treated with CHIR99021 did not. These findings

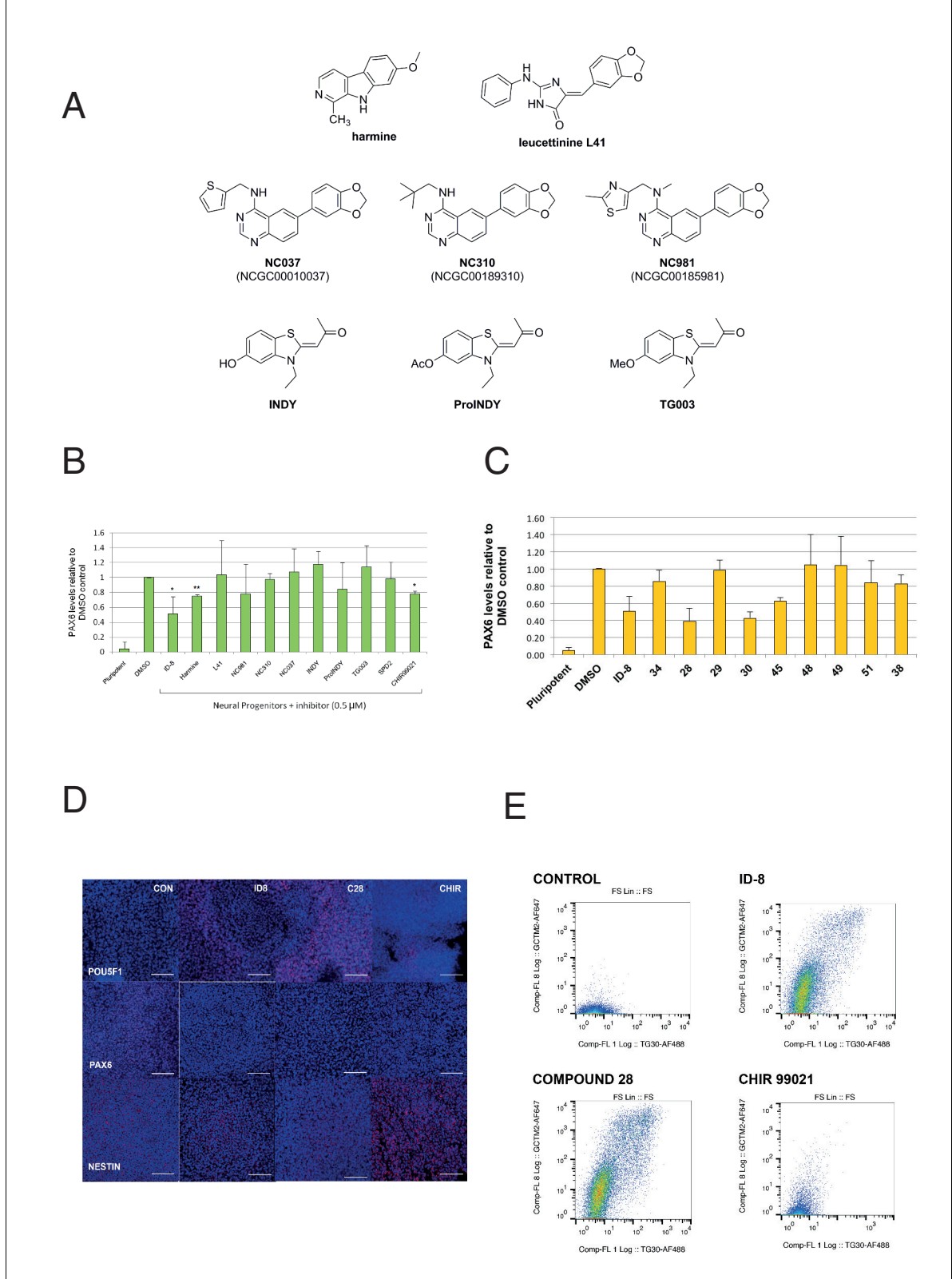

**Figure 3.** Effect of known DYRK1A and GSK3ß inhibitors and ID-8 analogues on induction of PAX6 positive cells by dual SMAD inhibition. (**A**) structures of known DYRK1A inhibitors studied. (**B**) percentage of PAX6 positive cells present at Day 16 in cultures of pluripotent cells, cultures of neural progenitors induced by dual SMAD inhibition (DMSO), and cultures subjected to neural induction in the presence of DYRK1A inhibitors at a dose of 0.5 µM. Error bars, SD; **p<0.05, *p<0.0.01. (**C**) percentage of PAX6 positive cells present at Day 16 in cultures of pluripotent cells, cultures of neural

*Figure 3 continued on next page*

*Figure 3 continued*

progenitors induced by dual SMAD inhibition (DMSO), or cultures subjected to neural induction in the presence of ID-8 analogues. All compounds were tested at 0.5 µM. Error bars, SD; **p<0.05, *p<0.0.01. (D) indirect immunofluorescence micrographs showing stem cell cultures subjected to dual SMAD inhibition alone or in combination with 5.0 µM ID-8, 5.0 µM compound **28**, or 3.0 µM CHIR 99021 and stained with antibodies to POU5F1, PAX6 or NESTIN (all red) and DAPI nuclear counterstain (dark blue). (E) flow cytometry analyses of stem cell surface marker (GCTM-2 antigen and TG30 anti-CD9 antibody) expression in cultures subjected to dual SMAD inhibition alone or in combination with 5.0 µM ID-8, 5.0 µM compound **28**, or 3.0 µM CHIR 99021. B and C, studies carried out with HES3 (PAX6$^{mCherry}$) cell line; D-E, WA09.

DOI: https://doi.org/10.7554/eLife.24502.009

suggest that CHIR99021 was acting in a manner different to ID-8 to block neural induction, most probably through activation of the Wnt signaling pathway to induce alternate neural states (more caudal multipotent progenitors or neural crest) in conjunction with inhibition of stem cell renewal by SMAD2/3 inhibition, as shown previously by several groups (*Chambers et al., 2016*; *Denham et al., 2012*; *Denham et al., 2015*; *Menendez et al., 2011*).

## ID-8 offsets effects of *DYRK1A* overexpression on neural specification of hESC and DYRK1A knockdown phenocopies effects of inhibitors

Off-target effects of ID-8 on DYRK1B, DYRK2, GSK3B, PIM1 or PIM3 or CLK1 or CLK4 might account for the inhibition of induction of neural differentiation by this compound. GSK3A, DYRK3, DYRK4, or LRRK2 are not expressed in hESC (www.stemformatics.org). The relationships between activity in kinase assays and activity of our panel of compounds in the neural induction bioassay described above would argue against a role for GSK3B, CLK1, CLK4, PIM1 or PIM3. To examine more directly the role of DYRK1A inhibition in the action of ID-8 on neural differentiation, we used a different model, namely a paradigm in which *DYRK1A* is specifically overexpressed in stem cells (*Figure 4a*). We used RNA guided gene activation mediated by the CRISPR system to switch on *DYRK1A* during induction of neural specification. gRNAs were designed to deliver a dCas9-VP64 activator construct to the *DYRK1A* regulatory region to enhance DYRK1A expression during induction of neural differentiation (*Figure 4a*). In this system, activation of *DYRK1A* (*Figure 4b*) during neural induction led to an increase in PAX6 expression that was offset by ID-8 or the other three inhibitors (compounds **28**, **30** and **45**, *Figure 4c*) active in the dual SMAD PAX6 induction assay described above. The DYRK1A inhibitors did not reduce the extent of induction of *DYRK1* by CRISPR activation (*Figure 4b*).

To further confirm the role of DYRK1A inhibition in blockade of neural specification, we used inducible shRNA to knockdown *DYRK1A* during neural specification by dual SMAD inhibition. WA09 hESC line was transduced with three separate doxycycline inducible *DYRK1A* shRNA constructs. Two out of three clones displayed strong knockdown of *DYRK1A* in the presence of doxycycline (*Figure 4d*). During neural specification, these two clones showed doxycycline inducible suppression of PAX6 expression (*Figure 4e*). Thus, *DYRK1A* knockdown phenocopies the effects of the inhibitors on PAX6 expression. We then confirmed the effects of inducible shRNA knockdown of DYRK1A in two more pluripotent stem cell lines, GENEA022 (the hESC line used in the DYRK1A activation study above) and C11(an hiPSC cell line). First we demonstrated that our active small molecule DYRK1A inhibitors could block the induction of PAX6 transcripts during neural induction by dual SMAD inhibition in WA09, GENEA022 and C11 cell lines (*Figure 4—figure supplement 1A*). Then we showed that shRNA knockdown of DYRK1A could phenocopy the effects of the inhibitors in GENEA022 and C11 cells (*Figure 4—figure supplement 1B,C*). Immunoblotting of detergent extracts of WA09 cells treated with scrambled control shRNA or DYRK1A shRNA in the presence of doxycycline confirmed significant reduction of DYRK1A protein levels by two of the constructs (*Figure 4—figure supplement 1D*).

## Modelling of ID-8/DYRK1A interaction

Eleven X-ray crystal structures are available for DYRK1A in complex with a range of ligands (*Figure 5—figure supplement 1*). In order to rationalize the structure-activity relationships of the ID-8 analogues, we sought to understand how these compounds bind to DYRK1A by molecular modelling. Seven reported structures deposited in the protein databank were superimposed using Maestro and the ligand binding sites were closely examined (*Figure 5a*). In the majority of cases the amino group of K188 and the amide nitrogen atom of L241 were involved in hydrogen bonding to the

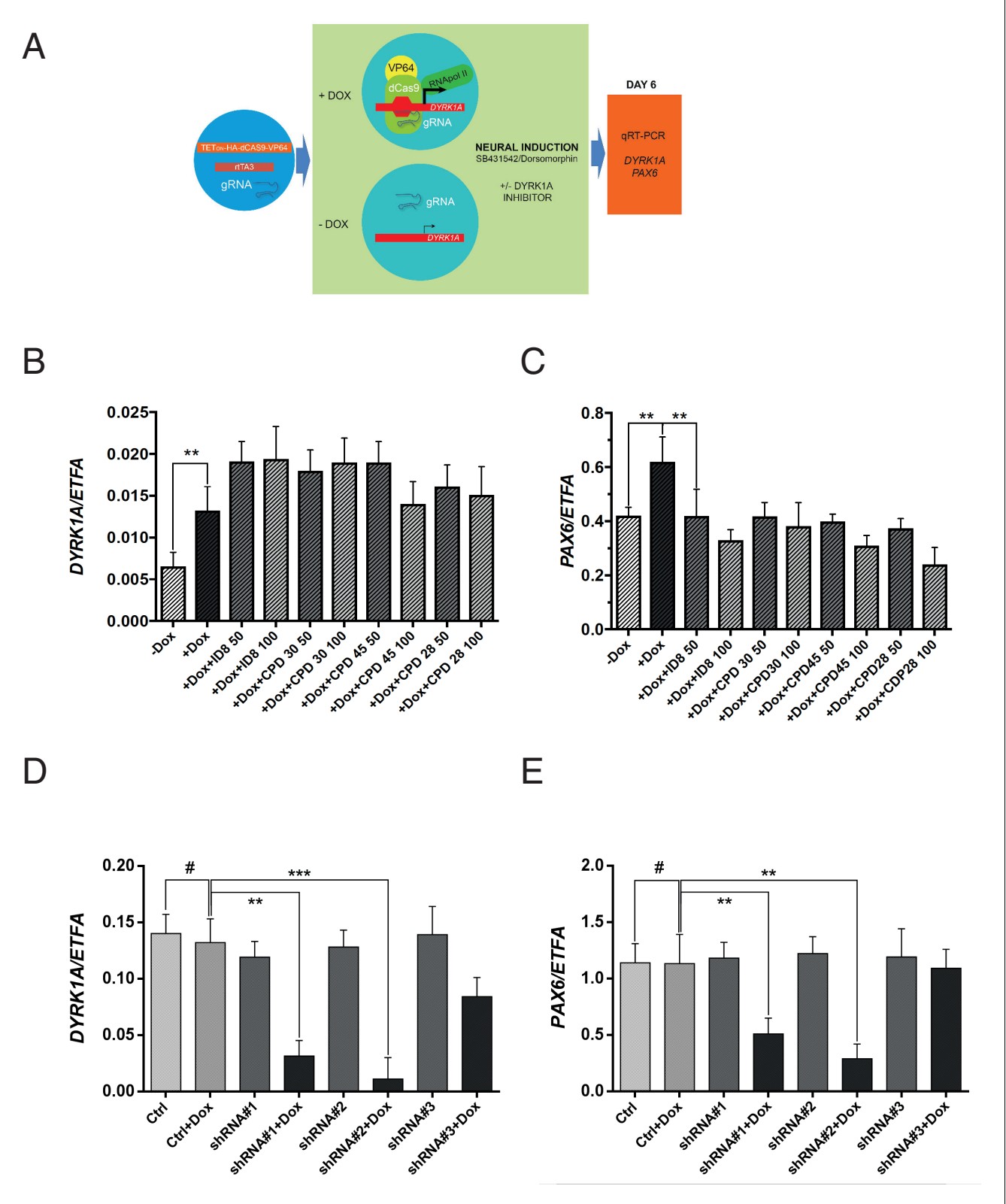

**Figure 4.** Effects of CRISPR activation and inducible knockdown of DYRK1A on PAX6 expression during neural specification by dual SMAD induction. (A) design of experiment to induce *DYRK1A*; (B) *DYRK1A* transcript levels following CRISPR activation in control cells or in presence of ID-8, or compounds **28**, **30**, or **45**. Error bars, SD; **p<0.05. (C) PAX6 transcripts following *DYRK1A* activation in control cells or cells treated with inhibitors at 50 or 100 nM. Error bars, SD; **p<0.05; all compounds produced significant reductions at all doses at p<0.05. (D) inducible shRNA knockdown of DYRK1A

*Figure 4 continued on next page*

*Figure 4 continued*

in WA09 hPSC. (**E**) effect of DYRK1A knockdown on PAX6 transcript levels following dual SMAD induction of neural specification. Result of triplicate experiments shown, mean ±SEM, **p<0.01, ***p<0.001. Studies in A and B were carried out with derivatives of hESC line WA09, those in C and D utilised derivatives of Genea022.

DOI: https://doi.org/10.7554/eLife.24502.010

The following figure supplement is available for figure 4:

**Figure supplement 1.** Inhibition of *PAX6* induction after dual SMAD inhibitor treatment by DYRK1A inhibitors or inducible *DYRK1A* shRNA in hESC and hiPSC cell lines.

DOI: https://doi.org/10.7554/eLife.24502.011

ligands. In these crystal structures containing diverse ligands, the ligands bind to both of these amino acids through polar interactions with either nitrogen or oxygen atoms. The average distance between both interacting atoms on the ligands was $8.3 \pm 0.2$ Å.

Using the crystal structure complexed with LDN-211898 (PDB code 5AIK) a set of docking experiments using ID-8 and compound **45** were undertaken. ID-8 docked in three alternative modes, one of which was very similar to the bound states of other DYRK1 ligands, as illustrated by overlay with the experimentally-determined LDN-211898 bound into 5AIK (*Figure 5b*). In this pose, the plane of the indole ring of ID-8 coincides with the aromatic sections of LDN-211898. ID-8 docks such that there is a hydrogen bond from the 6-hydroxyl to the amide group of L241, which is achieved from a slightly different direction to LDN-211898 by binding deeper into the active site, while the *para*-methoxy group of ID-8 is bound to E203 (adjacent to the amino group of K188). The distance between the two interacting atoms of ID-8 is 8.22 Å, similar to that of other DYRK1A inhibitors. Similarly, the amino group of K188 hydrogen bonds to the *para*-hydroxyl group of ID-8 by placing this hydroxyl group deep into the binding site. Compound **45** was able to dock into the binding site of 5AIK (*Figure 5c*) forming hydrogen bonds between the 6-hydroxyl and the amide nitrogen of L241 (2.97 Å) as well as the backbone carbonyl group of E239 (3.09 Å). The nitro group formed an interaction with the primary amino group of K188 (3.02 Å). The distance between the two interacting oxygen atoms of compound **45** is 7.78 Å. While compound **45** did not dock in the same mode as ID-8 it is easily be envisaged that this mode of binding is possible, as this compound possesses the same key polar interaction oxygens present in ID-8. A confounding factor encountered in docking both ID-8 and **45** is that the nitro group is more polar than the other two hydrogen bonding groups, and the docking algorithm is biased towards placing this group within the site to achieve polar interactions.

## Discussion

Here we present evidence that the indole compound ID-8 and a series of related molecules act to inhibit the neural specification of hESC through inhibition of DYRK1A. Our biochemical studies revealed that ID-8 is a relatively specific inhibitor of DYRK1A, and that the activity of a series of related compounds on neural specification most closely correlated to their ability to inhibit this enzyme. Further, in pluripotent stem cells specifically engineered to overexpress DYRK1A, neural specification induced by dual SMAD inhibition is enhanced, and this effect is offset by addition of ID-8 or other DYRK1A inhibitors. Finally, DYRK1A knockdown phenocopied the effect of the inhibitors on PAX6 expression during neural specification.

Our docking analysis provides a set of possible binding modes for ID-8 and compound **45**, which allow a rationalization of the potency of ID-8 analogues (Table 2) as inhibitors of DYRK1A. For example, the lack of inhibition for compounds **38** and **51** likely arise from the absence of a hydroxyl group at position 6 which therefore cannot interact with L241. Substitution at this position with a methyl group in compound **28** led to a small reduction in potency, while larger acetyl or benzyl in compounds **29**, **30** and **34** led to a marked reduction in potency. Compounds **48** and **49**, bearing methoxy groups oriented *ortho* or *meta*, respectively, cannot make ideal interactions with E203. Studies in man have implicated missense mutations affecting residues around the proposed binding site of ID-8 and compound **45** in the microcephaly syndrome associated with *DYRK1A* deficiency (*Ji et al., 2015*).

hPSC provide powerful models for understanding early human development. The biological actions of the DYRK1A inhibitors described here provide new insight into the effect of the absence

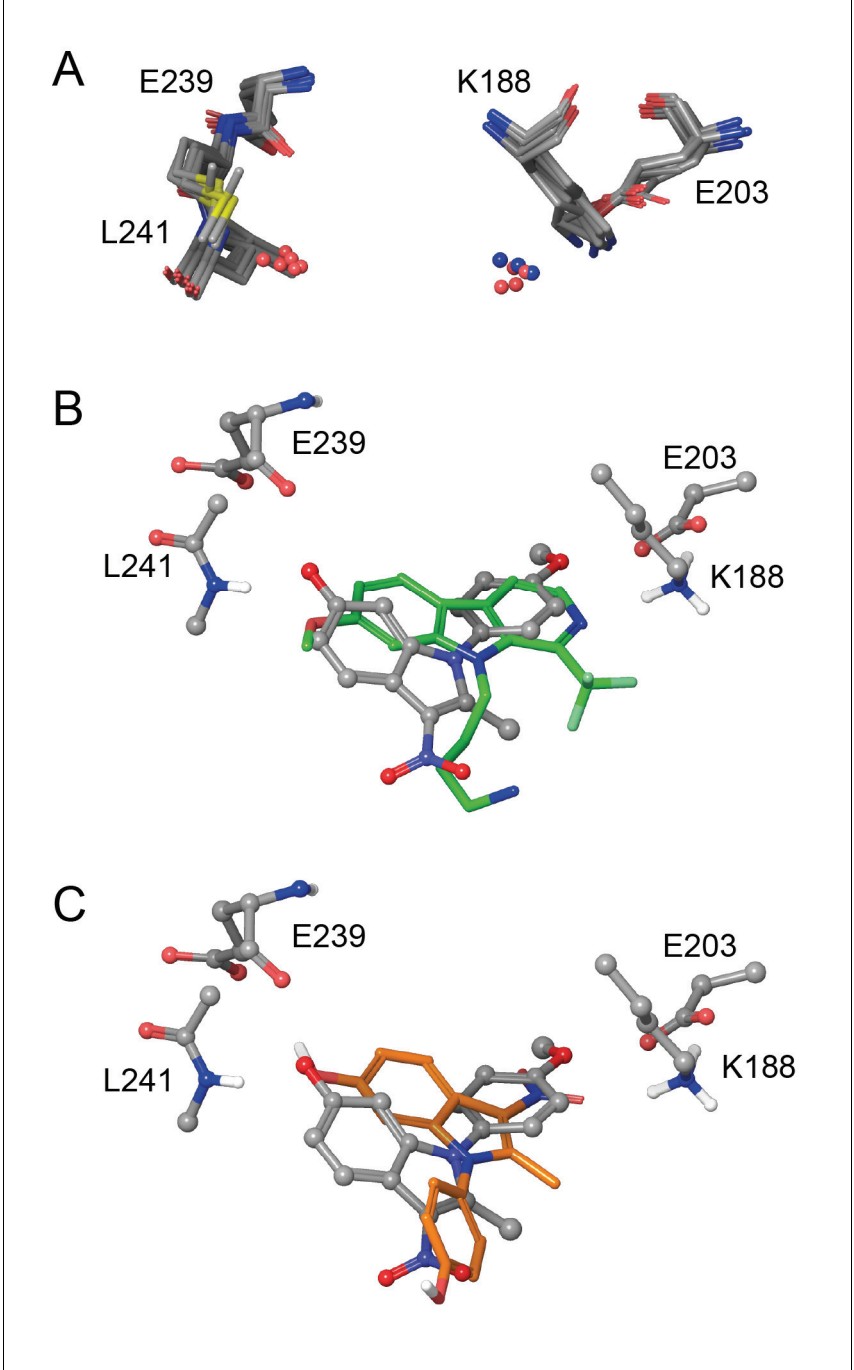

**Figure 5.** In silico modelling of binding of ID-8 and compound **45** to DYRK1A. (**A**) Overlay of seven ligand-bound X-ray structures of ligand-bound complexes of DYRK1A showing similar positioning of active site residues K188, E203, E239 and L241 (PDB codes: 5AIK, 3ANQ, 3ANR, 4YU2, 4MQ1, MQ2 and 4AZE). (**B**) Overlay of the docked pose of ID-8 (grey) with the experimentally-determined pose of LDN-211898 (green) with DYRK1A derived from the 5AIK structure. (**C**) Overlay of docked poses of ID-8 (grey) and compound **45** (orange). Maestro (version 2015–3) was used to dock ligands into 5AIK using the XP algorithm and the default settings were applied.

DOI: https://doi.org/10.7554/eLife.24502.012

The following figure supplement is available for figure 5:

**Figure supplement 1.** Summary of X-ray structures of DYRK1A and ligands bound.

DOI: https://doi.org/10.7554/eLife.24502.013

of this kinase on neural development in the embryo. In human or mouse, hemizygous loss of *DYRK1A* leads to microcephaly, a severe disorder of brain development. Some of the phenotypes of *Dyrk1a* knockout mouse, and haploinsufficient humans (*Bronicki et al., 2015*; *Courcet et al., 2012*; *Fotaki et al., 2002*; *Ji et al., 2015*; *Møller et al., 2008*), could be explained by an action at a very early stage of lineage specification in the embryo. Our previous work showed that ID-8 treatment of hESC blocked the ability of WNT3A to induce expression of mesoderm or endoderm genes (*Hasegawa et al., 2012*). This study reveals that DYRK1A inhibition specifically interferes with the early specification of hESC to neural fates. Together, these findings suggest that *DYRK1A* loss might exert a profound effect on endowment of the three germ layer lineages in homozygous or heterozygous null embryos. *Dyrk1a* homozygous null mice die at E10.5, with growth delay in multiple organ systems, including heart, brain, liver and branchial arches. Humans heterozygous for loss or mutation of *DYRK1A* display severe microcephaly, but also commonly show developmental delay and intrauterine growth retardation. It is possible that failure to allocate sufficient numbers of pluripotent progenitors into the three germ layer lineages could result in generalised developmental deficiencies and delay, along with severe central nervous defects. The neural specification assay here is suitable for rapid screening of the effects of DYRK1A inhibition on neural development in human cells, and could aid in development of novel inhibitors, as could the CRISPR mediated DYRK1A induction assay, which mimics the gene dosage effect of DYRK1A overexpression in Down syndrome. Indeed, the results of this assay suggest that DYRK1A inhibitors could be used to ameliorate neural phenotypes in Down syndrome, though clearly the dose of the agents would need to be controlled rather precisely to avoid excessive reduction in enzyme activity.

Our observations are in contrast to previously reported actions of DYRK1A inhibition in mouse embryonic stem cells, where treatment with epigallocatechin-gallate caused reduced expression of pluripotency genes and enhanced expression of mesodermal and endodermal lineage specific genes (*Canzonetta et al., 2008*). However, epigallocatechin-gallate is a broad spectrum kinase inhibitor that lacks specificity for DYRK1A, and its effects on mouse ES cells could be attributable to interference with other pathways, or to species differences in stem cell regulation.

The actions of DYRK1A inhibitors on human pluripotent stem cells provide important insight into the role of this kinase in stem cell regulation. ID-8 treatment did not affect expression of stem cell markers in cultures maintained under conditions that support self-renewal. Neither did the compound affect proliferation of stem cells maintained under these conditions. We showed previously that ID-8 could promote colony formation of hESC, but this effect was modest, except in combination with WNT3A. The strongest effect of treatment of hESC with ID-8 observed in our previous work was the inhibition of induced differentiation. In cells treated with WNT3A (*Hasegawa et al., 2012*), WNT3A induction of mesodermal and endodermal genes was strikingly attenuated by ID-8. Here we showed that the powerful neural induction using dual SMAD inhibition could be offset by treatment of stem cells with ID-8.

Careful inspection of flow cytometry profiles after ID-8 treatment revealed that the response of the cell population to DYRK1A inhibition was not homogenous. In fact, a large proportion of cells treated with dual SMAD inhibition with or without ID-8 lost stem cell surface markers. However, in ID-8 treated cultures, the cellular subset bearing the highest levels of stem cell markers appeared to remain intact, and indeed was proportionally expanded 140 fold in differentiating cultures relative to other subpopulations under the condition of DYRK1A inhibition. We have previously shown that this cellular subset represents the self-renewing component of hESC cultures, and that it has molecular properties distinct to the bulk of the stem cell population (*Hough et al., 2014*). Recently we have observed that this subpopulation of hESC has an unusual cell cycle, with a short or non-existent G1 phase compared to the remainder of the population (unpublished). It is known that hPSC become competent to respond to differentiation signals in G1 phase of the cycle (*Boward et al., 2016*). Our results suggest that DYRK1A inhibition by ID-8 prevents the self-renewing subpopulation from acquiring competence to respond to induction of differentiation, rendering it refractory to strong extrinsic inductive signals. Thus, transition to a differentiation permissive state may require activation of DYRK1A, an effect that may be actuated through the lengthening of G1 in the self-renewing population. Inhibition of DYRK1A could prevent the lengthening of the G1 phase and thereby block progression of the self-renewing hESC subpopulation towards neural specification. Further study of DYRK1A will provide additional insight into its role in early development, and the mechanism

whereby its activity modulates the competence of hESC to respond to inducers of lineage specification.

## Materials and methods

### DYRK1A inhibitors and other compounds

LDN193189 (Stemgent, Lexington, MA, #130-096-226), **SB431542** (Stemgent, #130-097-448), **CHIR99021** (Stemgent, #130-095-555) and harmine hydrochloride (Sigma-Aldrich, St. Louis, MO, #SMB00461) were purchased from commercial suppliers. All other inhibitors were synthesised as described in the Supplemental Methods. All compounds were solid powders, stable at room temperature. Compounds were dissolved in DMSO to achieve desired stock concentrations, and were aliquoted for long-term storage at $-20$ or $-80°C$. Stock concentrations were diluted further in DMSO, to desired concentration for adding to cell culture, such that DMSO concentration in cell culture media did not exceed 0.15%.

### Kinase inhibition assays

Kinase inhibition assays were performed by Reaction Biology (www.reactionbiology.com). Each kinase was produced as a recombinant protein and used a matched peptide substrate. The reaction buffer was 20 mM Hepes (pH 7.5), 10 mM $MgCl_2$, 1 mM EGTA, 0.02% Brij35, 0.02 mg/ml BSA, 0.1 mM $Na_3VO_4$, 2 mM DTT, 1% DMSO. Radioassays were performed using 10 µM ATP and 10 µCi/µl [γ-$^{33}$P]ATP. Enzyme concentrations were 0.2–50 nM, and peptide substrate concentration was 20 µM. Reactions were incubated for 2 hr at room temperature, then spotted onto P81 ion exchange paper, and product detected by radiometry. For initial screening, candidate inhibitors were included at 10 µM, and activity was calculated as percentage inhibition relative to the control with no inhibitor. For determination of $IC_{50}$ values, inhibitors were included into kinase reactions at 10 concentrations with 3-fold serial dilutions starting at 30 µM.

### Cell lines

WA09 and HES3 (*PAX6^mCherry*)cell lines were maintained by and provided by the Stem Cell Core Facility of Stem Cells Australia. The knock-in reporter HES3 (*PAX6^mCherry*) cells were developed as described (below). hESC lines WA09 (obtained from WiCell (WA09 MEF platform), GENEA022 (provided by Genea Biocells), and hiPSC line C11 (*Briggs et al., 2013*) were maintained at the University of Queensland laboratories on fibroblast feeder cell support in medium supplemented with Knockout Serum Replacer and FGF-2 (Invitrogen/Thermofisher Scientific, Waltham, MA), with daily media changes. For propagation in defined media, WA9 cells were adapted from culture on mouse embryo fibroblasts (mechanically dissociated organ cultures) at passage 74 or 87, and maintained thereafter on hES-qualified Matrigel in mTesR1 (STEMCELL Technologies, Vancouver, BC, Canada) medium using dispase to disaggregate the cells. HES3(*PAX6^mCherry*) were cultured on hES-qualified Matrigel in mTesR1 medium. Medium was changed daily, and cells were passaged with dispase. Cell line identity was confirmed by STR profiling.

### Generation of the PAX6 targeting vector

DNA fragments representing the 5' and 3' homology arms were derived from bacterial artificial chromosome clones (BAC) RP11-26B16 and RP11-307I15 (Roswell Park Cancer Institute), which included sequences spanning the PAX6 stop codon. The PAX6 targeting vector was assembled using a combination of recombineering and standard cloning procedures. Sequences representing the 3.6 Kb 5' homology arm were amplified by PCR and subsequently cloned into a vector containing a PGKNeo selection cassette and sequences corresponding to the 7.8 Kb 3' homology arm, the latter having been previously introduced into this plasmid using recombineering. This vector was then linearised at an Mlu1 site at the junction of the 5' homology arm and the PGKNeo selection cassette, enabling the sequential introduction of a T2A sequence that replaced the PAX6 stop codon and, an mCherry-IRES-puromycin cassette. Prior to electroporation, the final plasmid was digested with Pac1 and AsiS1 to release the targeting vector as depicted in *Figure 2—figure supplement 2a*.

## Generation of the PAX6-mCherry reporter line

The HES3 cell line (*Richards et al., 2002*) was electroporated with the PAX6 targeting vector using standard conditions (*Costa et al., 2007*). Single G418 resistant colonies were screened for correct integration of the targeting vector using PCR primers that corresponded to sequences within the PGKneo cassette and genomic sequences 3' of the 3' end of the 3' PAX6 homology arm: neo4 (cgatgcctgcttgccgaatatcatg) & 39783R (gccgattcctgagcctttcatacc). Two positive clones identified in this initial screen were then validated by repeating the PCR analysis performed in the original screen and by performing a second PCR analysis that confirmed correct recombination of the vector at the 5' end. This second PCR, which utilised the primer pairs cherryrev2 (ccatgttatcctcctcgcccttgc) and 27749F (gctaacctgtcccacctgatttcc), generated the predicted product of 8.6 Kb. The result of this confirmatory PCR analysis is shown in *Figure 2—figure supplement 2b*. These positive clones were expanded and subsequently transfected with a cre recombinase expression vector as described previously (*Davis et al., 2008*), resulting in excision of the PGKNeo cassette. Successful removal of this cassette was confirmed using PCR analysis. Subclones of each primary clone were then submitted for karyotype analysis, performed by the Cytogenetics Department at Southern Cross Pathology, Monash Medical Centre. A single subclone, PAX6 # 85.2 with a normal female karyotype was then chosen for further analysis.

## Neural induction assay

The neural induction assay involved the directed differentiation of hESC to neural progenitor cells, and is based on the dual-SMAD inhibition methods by *Chambers et al. (2009)*. HES3(*PAX6^mCherry*) or WA09 cells in mTeSR1 were passaged using 1 mg/ml dispase onto Matrigel coated 3 cm dishes and cultured in mTesR1. The cells were grown to minimum 30% confluency prior to supplementation with 5 µM SB431542 and 100 nM LDN193189, in the presence or absence of kinase inhibitors diluted in DMSO, or DMSO vehicle only for the controls, with a total DMSO concentration of 0.15% in each dish. Media was changed every 2 days, with continual inhibitor supplementation. At day 7, the cells were transferred into Neurobasal media (Invitrogen/Thermofisher) with FGF2 and EGF, continuing kinase inhibitor supplementation, without LDN/SB inhibitors. At day 9, cells were mechanically lifted/ scraped from the bottom of the dish, dissociated to clumps by pipetting, and passaged onto fresh Matrigel coated 3 cm dishes in Neurobasal medium supplemented with 5 µM Rho kinase inhibitor Y-27632, achieving approximately 30% confluency immediately following passage. At day 11, Rho kinase inhibitor was withdrawn, and cells were maintained in Neurobasal media with continual supplementation of DYRK inhibitors with media changes every 2 days, until the cells were fixed for immunofluorescence or harvested for flow cytometry. For statistical analysis, P values were calculated using a paired two-tailed Student's t-test with correction for multiple comparisons using the Benjamin-Hochberg procedure with a false discovery rate of 0.075. All experiments were repeated 2–3 times with three technical replicates in each data point.

## Measurement of pluripotency marker expression, proliferation and apoptosis

Flow cytometry measurement of the expression of the pluripotent stem cell surface markers GCTM2 and CD9, and indirect immunofluorescence microscopy for POU5F1 and nestin on paraformaldehyde fixed adherent cell cultures, were performed as previously described (*Hough et al., 2014*). The cell cycle distribution of control and ID-8 treated pluripotent WA09 cells was analysed using the Click-IT EdU Flow Cytometry Kit AF488 (Invitrogen/Thermofisher, #C-10425). Apoptosis of control and ID-8 treated pluripotent WA09 cells was analysed using the Abcam In situ BrdU-Red DNA Fragmentation (TUNEL) Assay Kit (Abcam, Cambridge, MA). Both kits were used according to the manufacturers' instructions. All experiments were repeated two to three times.

## Effect of indole compounds on induction of neural specification driven by DYRK1a overexpression

To identify gRNAs capable of driving CRISPRa activation of the *DYRK1A* gene, a number of unique 20mers in region between −250 to −50 bps relative to the main transcription start site were cloned in the pX462 gRNA and dCas9-VP64-expression vector and tested by transfection into the 293FT human cells, with subsequent qPCR analysis of *DYRK1A* expression. Generation of stable cell lines

was achieved by transduction of the 2 selected best-performing gRNA (gRNA #1 CCGGCAAA TACCGCAGTCCC; gRNA#2, CGCTGGAACCGCGAGCCGAG)-expressing lentiviruses (pLKO.1-sgRNA-Neo) into the recipient pre-validated hESC Genea022 line expressing HA-dCas9-VP64 in a doxycycline-inducible fashion. Neural differentiation, used as a backdrop for the inhibitor assay, was performed for 6 days in KSR medium supplemented with 10 µM SB431542 and 0.5 µM LDN193189 and devoid of pluripotency-maintenance growth factors. Control cultures grown ±doxycyline, and cultures treated with inhibitor compounds, contained 0.1% DMSO for the duration of the assay. For statistical analysis, P values were calculated using a paired two-tailed Student's t-test with correction for multiple comparisons using the Benjamin-Hochberg procedure with a false discovery rate of 0.075. All experiments were repeated 2–3 times with three technical replicates per data point.

## DYRK1A knockdown studies

For an inducible knockdown of the DYRK1A expression, we transduced human pluripotent stem cell cultures (WA09, GENEA022, and C11) with VSV-G-pseudotyped lentiviral particles: one control (non-targeting) and three DYRK1A ORF-targeting shRNAs in doxycycline-inducible SMARTvectors (GE Healthcare/Dharmacon, product # DHA-VSC11653 and V3SH11255). The virus was packaged using the 293FT cell line (LifeTechnologies/Invitrogen) transfected with a second-generation lentiviral packaging plasmid psPAX2, a gift from Didier Trono (Addgene plasmid # 12260). Selection for cells harboring expressable proviral integrants was started 5 days after transduction with puromycin at 2 µg/mL. DYRK1A knockdown was induced by treatment with doxycycline at 0.5 µg/mL for 3 days prior to, and during the 6 days of, the dual SMAD inhibition-driven neural differentiation. Effects of small molecule DYRK1A inhibitors and shRNA on *DYRK1A* and *PAX6* transcript induction during neural specification were assessed in the cell lines used for knockdown studies at Day 16 in the protocol described above. *PAX6* transcript levels were measured by qPCR on total RNA (0.5 µg)-derived cDNA (using iScript polyA and random-priming cDNA synthesis system, Bio-Rad) extracted from pluripotent cells subjected to a standard (dual SMAD inhibition) neuroepithelial differentiation in presence of the indicated compounds for 16 days. Detergent extracts of shRNA treated cells were separated on denaturing and reducing 12% tris-glycine-sodium dodecyl sulphate acrylamide gels. The samples were prepared using a protease and phosphatase-supplemented mix of RIPA and 2x Laemmli buffers (Sigma-Aldrich). Following gel transfer to polyvinylidene difluoride membranes, the blots were probed using primary antibodies against DYRK1A (D30C10 Rabbit mAb #8765, Cell Signalling Technologies,U.S.A.) and GAPDH (14C10 Rabbit mAb #2118, CST), and a secondary anti-rabbit IgG, HRP-linked Antibody (#7074, CST). Visualisation was performed using Clarity ECL substrate, and densitometric quantification performed using ImageLab4.1 software (Bio-Rad, U.S.A.).

## Structural modelling

Eleven crystal structures are available of DYRK1A with bound ligands. These were superimposed using Maestro and an analysis of the binding sites was carried out. Maestro (version 2015–3) was used to dock ligands into 5AIK using the XP algorithm and the default settings were applied. The protein was prepared using the protein preparation wizard.

## Acknowledgements

This work was supported by the University of Melbourne, the Australian Research Council (SR1101002). Work in the laboratories of EGS and AGE was supported by grants from the Australian Stem Cell Centre, Stem Cells Australia, and the National Health and Medical Research Council (NHMRC) of Australia. AGE and EGS are Senior Research Fellows of the NHMRC.

## Additional information

### Competing interests

Martin Pera: Reviewing editor, *eLife*. The other authors declare that no competing interests exist.

## Funding

| Funder | Grant reference number | Author |
|---|---|---|
| Australian Research Council | Special Research Initiative SR1101002 | Dmitry A Ovchinnikov<br>Andrew G Elefanty<br>Edouard G Stanley<br>Ernst J Wolvetang<br>Martin Pera |
| National Health and Medical Research Council | Senior Research Fellowship | Andrew G Elefanty<br>Edouard G Stanley |
| University of Melbourne | Strategic APA | Stephanie F Bellmaine<br>Spencer J Williams<br>Martin Pera |

The funders had no role in study design, data collection and interpretation, or the decision to submit the work for publication.

## Author contributions

Stephanie F Bellmaine, Conceptualization, Investigation, Visualization, Methodology, Writing—original draft, Writing—review and editing; Dmitry A Ovchinnikov, Conceptualization, Investigation, Visualization, Methodology, Writing—review and editing; David T Manallack, Conceptualization, Data curation, Formal analysis, Investigation, Visualization, Methodology, Writing—review and editing; Claire E Cuddy, Formal analysis, Validation, Investigation; Andrew G Elefanty, Resources, Methodology, Writing—review and editing; Edouard G Stanley, Resources, Visualization, Methodology, Writing—original draft, Writing—review and editing; Ernst J Wolvetang, Conceptualization, Resources, Methodology, Writing—review and editing; Spencer J Williams, Conceptualization, Resources, Supervision, Funding acquisition, Visualization, Methodology, Writing—original draft, Writing—review and editing; Martin Pera, Conceptualization, Resources, Formal analysis, Supervision, Funding acquisition, Writing—original draft, Project administration, Writing—review and editing

## Author ORCIDs

Ernst J Wolvetang (iD) https://orcid.org/0000-0002-2146-6614
Martin Pera (iD) http://orcid.org/0000-0001-6239-0428

## Decision letter and Author response

Decision letter https://doi.org/10.7554/eLife.24502.018
Author response https://doi.org/10.7554/eLife.24502.019

# Additional files

## Supplementary files

• Supplementary file 1. Activity of ID-8 against a range of 338 protein kinases. Assays were performed as described in Materials and methods. The table shows percentage of activity relative to control at a 10 µM concentration of ID-8, along with $IC_{50}$ (M) values for positive control compounds.
DOI: https://doi.org/10.7554/eLife.24502.014

• Supplementary file 2. Chemical syntheses. The text describes general experimental procedures and details of chemical syntheses and physical properties of the novel compounds used in this study.
DOI: https://doi.org/10.7554/eLife.24502.015

• Supplementary file 3. Kinase inhibition of ID-8 relative to other known DYRK1A inhibitors. Table shows $IC_{50}$ (nM) or percentage inhibition values for compounds against a series of CMGC kinase family members. Values are from this study (ID-8, see Materials and methods) or the literature cited.
DOI: https://doi.org/10.7554/eLife.24502.016

• Transparent reporting form
DOI: https://doi.org/10.7554/eLife.24502.017

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
