## [Decision Letter]

Thank you for submitting your article "Inhibition of DYRK1A disrupts neural lineage specification in human pluripotent stem cells" for consideration by *eLife*. Your article has been reviewed by two peer reviewers, and the evaluation has been overseen by a Reviewing Editor and Marianne Bronner as the Senior Editor. The reviewers have opted to remain anonymous.

The reviewers have discussed the reviews with one another and the Reviewing Editor has drafted this decision to help you prepare a revised submission.

Summary:

We have now received two reviews on your manuscript "Inhibition of DYRK1A Disrupts Neural Lineage Specification in Human Pluripotent Stem Cells". While the second reviewer is more enthusiastic than the first, there seems to be general agreement that the manuscript is potentially publishable in *eLife* after additional experimental work is done.

Essential Revisions:

Since the paper focuses heavily on ID-8 and its analogues, it is very important to show definitively that the action of the compounds is truly via inhibiting DYRK1a. It is not sufficient to show that particular compounds have activities in cell-free systems that appear to translate into their effects on cells. The authors themselves state that some of the enzymes that can be inhibited by the compounds are not even expressed in huES cells. Moreover, it may be that the potency differences observed across a series of compounds can be ascribed to variability in cellular permeability. Thus, as requested by both reviewers, it is key that the authors provide more evidence that the compounds act on target cells at their effective concentrations by inhibiting DYRK1A and not, for example, GSK3β. Furthermore, as requested by both reviewers, more information should be provided as to the details of compound testing in cell-free assays and on the ES cells. These items are addressed in detail by Reviewer 1 (points 1-3); Reviewer 2 (point 3b). In addition, please make clear what conclusions you are drawing from Figure 3. Along those lines, both reviewers also request that you show convincingly using genome modification methods that DYRK1a reduction or deletion phenocopies the effects of your compounds.

In addition, both reviewers would like you to show that the compounds have similar effects on more than 1 huES line. Reviewer 2 has requested a confirmatory study (point 2) that seems important to carry out since it would provide strong support for the mechanisms that you propose.

*Reviewer #1:*

This paper described the optimization of a small set of ID-8 like compounds as specific inhibitors for DYRK1A and DYRK1B that bind to the ATP binding pocket. Using a computational model, the authors rationalized the SAR within the series on DYRK1A. Using these chemical tools, together with the cell line reporting the level of PAX6 expression, as well as CRISPR-mediated gene activation, the authors argue that the inhibition of DYRK1A prevents the neural specification of huES cells. The ID-8 is previously proposed to be the potential DYRK inhibitor, although not as characterized, and the molecular mechanisms underlying DYRK1A's effect is not characterized in this report.

1) The methods for the kinase assays in Table 1 and 2 should be detailed, including the source of kinase protein preparation, substrate peptides, K_m_ for each peptide substrate, and ATP? Is each assay performed at Km for substrate and K_m_ for ATP? As the IC_50_s of the chemical compounds depend on the kinase assay condition, such information must be provided, and critical in evaluating the results for specificity. Why the discrepancy between Table 1, and Table 2 in terms of the IC_50_s of ID8 against DYRK1A (0.054 vs. 0.104μM), GSK3b (0.45μM vs. 0.15μM)?

2) For the evaluation of cellular activities of ID8 series compounds, dose responses of key compounds in this assay, should be included in the Table 2. Cellular toxicity needs to be addressed. At 5 μm, does ID-8 cause any reduction in the number of cultured cells (Figure 2, right panel)? CRISPR-mediated conditional deletion of DYRK1A would be a direct and conclusive genetic evidence to demonstrate that DYRK1A is necessary to support the neural specification, and should be included in the argument.

3) It is not conclusive whether GSK3b underlies the effect of ID-8 on neural specification, as CHIR99021 has an effect on PAX reporter assay. CHIR99021 has a different profile across kinome, and the noticed difference on Oct-4 expression may be derived from off-target effects of CHIR99021. Providing a GSK3b dependent cellular readout with ID-8 compounds in these cells, such as p-tau, or p-CRMP, could help answer wither ID-8 compounds hit GSK3b in these cells.

4) The author showed that ID-8 compounds blocked the neural specification, most likely through inhibiting DYRK1A. Previously, it is shown that ID-8 promotes the stem-ness of huES cells by promoting self-renewal (Hasegawa et al., 2012). Are the observations in this report merely reflecting the fact that the huES cells in the presence of ID8 have more stem-ness, hence resistance to be induced for any lineage specialization, or that DYRK1A is a key switch for neural lineage specification? Can huES cells be induced into neurons, in the presence of ID8, by overexpressing neurogenin2 that bypass the neuroprogenitor stage (Sudhof lab, 2013 Neuron)? In either case, the molecular targets need to be identified/discussed to provide a mechanism for such observation, are EGFR, FGF, p53 etc. involved in such a functional role (PMID 20237271; PMID 18771760; Ferron et al., 2010).

*Reviewer #2:*

In their manuscript entitled "Inhibition of DYRK1A Disrupts Neural Lineage Specification in Human Pluripotent Stem Cells," Bellmaine et al. demonstrate that the addition of ID-8, and similar chemical inhibitors of DYRK1A, prevents hESCs from differentiating to NPCs. This is a very clear and straightforward report. With what I hope are some simple additional experiments, this manuscript could be suitable for publication in *eLife*.

1) Inter-hESC line variability – in the Materials and methods, the authors describe using each of a HES3-knockin-PAX6 reporter, H9 and WA9 hESC lines. It would be helpful if they clarified which lines were used in which figures. Moreover, the manuscript would be improved if the effect of ID-8 was directly compared across all three hESC lines in a single figure panel (i.e. Pax6 qPCR or AB-based FACS).

2) NPCs: The authors suggest that ID-8 treatment is preventing hESCs from specifying to an NPC fate, supported by data showing ID8 doesn't change hESC replication or pluripotency markers (Figure 2—figure supplement 1). To confirm that this is a patterning defect, rather than a replication, death or differentiation phenotype, would it be possible to FACS purify remaining Pax6-GFP positive cells following ID8 treatment and assess their capacity to replicate and undergo neuronal differentiation (either with or without ID8 present)? Specifically, I am curious whether the remaining NPCs generate both neurons and astrocytes, and if neurons, both excitatory and inhibitory.

3) Disease relevance: The authors make repeated note of the presumed role of DYRK1A in neurodevelopmental disease, both in their Introduction and the Discussion, but fail to discuss any similarities between DYRK1A+\- hiPSC phenotypes and ID8 effects. Can they phenocopy DYRK1A+\- phenotypes across an ID8 dose-response curve? Even more important, can they rescue developmental effects in DS hiPSCs across an ID8 dose-response curve? (I understand that this would be the most difficult experiment of all the ones I have suggested, and that DS hiPSCs may not be available to this group – it is just the most interesting question I had in reading this manuscript.)

a) Given that the authors demonstrated an ability to prevent neural differentiation with ID8, even when up-regulating DYRK1A, can they similarly impair neural differentiation by knocking down DYRK1A with dCas9-KRAB? This would confirm the specificity of the ID8 activity.

b) Similarly, DYRK1A is a member of the dual-specificity tyrosine phosphorylation-regulated kinase family. It must have some known downstream targets that the authors can query by western blot, etc., to confirm the specificity of ID-8. If this was done in previous publications, it should be noted and cited.

4) Technology relevance: Does the addition of ID8 improve the efficacy/yield of non-neuronal (i.e. cardiac, muscle, endoderm) differentiation protocols? Are there any known activators of DYRK1A that could be used to improve neuronal differentiation protocols?

[Editors' note: further revisions were requested prior to acceptance, as described below.]

Thank you for resubmitting your work entitled "Inhibition of DYRK1A disrupts neural lineage specification in human pluripotent stem cells" for further consideration at *eLife*. Your revised article has been favorably evaluated by Marianne Bronner (Senior Editor), a Reviewing Editor, and two reviewers.

The manuscript has been improved but there are some remaining issues that need to be addressed before acceptance, as outlined below:

After additional discussion with the reviewers, we reached the conclusion that you should still carry out the experiment highlighted as reviewer 2's point #1 (repeating your main assay using 2 additional lines). We believe that this experiment is straightforward to do and will be important in convincing readers of the robustness of your observations.

*Reviewer #2:*

In their revised manuscript, Bellmaine et al. now better demonstrate that inhibition of DYRK1A prevents hESCs from differentiating to NPCs. The authors were reasonably responsive to my concerns, particularly in adding a genetic inhibition of DYRK1A to complement the pharmacological studies. Nonetheless, two simple experiments remain.

1) Inter-hESC line variability – the authors did a great job clarifying which hESC lines were used in each figure panel. Nonetheless, I still think it would be helpful to compare all three hESC lines across at least a single assay (i.e. Pax6 qPCR or FACS), in order to assess how robust the effects are across hESC lines.

2) NPCs: The authors suggest that ID-8 treatment is preventing hESCs from specifying to an NPC fate. It would be interesting to collect the remaining Pax6-GFP positive cells following ID8 treatment (or shRNA knockdown) and assess either their gene expression and/or their capacity to replicate and undergo neuronal differentiation (either with or without ID8 present)? Specifically, I am curious whether the remaining NPCs generate both neurons and astrocytes, and if neurons, both excitatory and inhibitory.

---

## [Author Response]

Essential Revisions:Since the paper focuses heavily on ID-8 and its analogues, it is very important to show definitively that the action of the compounds is truly via inhibiting DYRK1a. It is not sufficient to show that particular compounds have activities in cell-free systems that appear to translate into their effects on cells. The authors themselves state that some of the enzymes that can be inhibited by the compounds are not even expressed in huES cells. Moreover, it may be that the potency differences observed across a series of compounds can be ascribed to variability in cellular permeability. Thus, as requested by both reviewers, it is key that the authors provide more evidence that the compounds act on target cells at their effective concentrations by inhibiting DYRK1A and not, for example, GSK3β.

We agree with this point and provide new data and some additional background information to address these concerns. First, with respect to inhibition of GSK3b as a potential mechanism (the active compounds in our study showed some inhibitory activity towards this kinase), we showed in the first submission that the potent and selective GSK3b inhibitor CHIR 99021 also reduced PAX6 induction following dual SMAD inhibition, similar to DYRK1A inhibitors. However, we argued that the action of CHIR 99021 was fundamentally different to that of DYRK1A inhibitors, because treatment with the latter agent but not the former resulted in retention of a pluripotent phenotype. We based this claim on indirect immunofluorescence staining for nestin and POU5F1 following CHIR 99021 treatment shown in Figure 3 (compare to Figure 2 with ID-8). We grant that this evidence on its own is fairly weak, so we now provide further support for this interpretation. First we cite previous studies showing that inhibition of GSK3b in combination with SMAD inhibition in hPSC indeed blocks PAX6 induction but directs cells towards other neural fates-neural crest or more caudal neural precursors (1-4) subsection “Effects of known DYRK1A and GSK3β inhibitors and ID-8 analogues on neural specification of hESC”, last paragraph. These studies (which we should have noted in the first submission) are consistent with our findings on CHIR 99021 but fundamentally different to our results with ID-8. We also provide new immunocytochemistry and flow cytometry data which confirm results in Figure 2 and show that while co-treatment of cells with ID-8 (or another DYRK1A inhibitor, compound 28) and dual SMAD inhibition results in retention of pluripotent phenotype by a significant fraction of cells, co-treatment with CHIR 99021 plus dual SMAD inhibitors results in complete loss of stem cell marker expression across the population, similar to dual SMAD inhibition alone subsection “Effects of known DYRK1A and GSK3β inhibitors and ID-8 analogues on neural specification of Hesc”, last paragraph, and new Figure 3. See also the response to point 4 below.

Furthermore, as requested by both reviewers, more information should be provided as to the details of compound testing in cell-free assays and on the ES cells. These items are addressed in detail by Reviewer 1 (points 1-3); Reviewer 2 (point 3b).

We now include the details of the assays in the Materials and methods subsection “Kinase inhibition assays”. See also responses to reviewers.

In addition, please make clear what conclusions you are drawing from Figure 3.

We apologise that the point of this figure was not explained more clearly. This figure shows that some known (albeit less specific) inhibitors of DYRK1A also block neural lineage specification, that treatment with a GSK3B inhibitor CHRI 99021 has this effect but does not result in retention of pluripotency, and that other potent and specific DYRK1A inhibitors that we synthesised block neural lineage specification. We now include a subheading to introduce the presentation of the data in Figure 3 new figure title which we hope will clarify the objective of the studies therein, subsection “Effects of known DYRK1A and GSK3β inhibitors and ID-8 analogues on neural specification of hESC” and Figure 3 legend “Effect of known DYRK1A and GSK3ß inhibitors and ID-8 analogues on induction 588 of PAX6 positive cells by dual SMAD inhibition”.

Along those lines, both reviewers also request that you show convincingly using genome modification methods that DYRK1a reduction or deletion phenocopies the effects of your compounds.

We include new data showing that inducible shRNA knockdown of DYRK1A indeed blocks PAX6 induction by dual SMAD inhibition (Summary; Introduction, last paragraph; subsection “ID-8 offsets effects of DYRK1A overexpression on neural specification of hESC and 256 DYRK1A knockdown phenocopies effects of inhibitors”; Discussion, first paragraph; subsection “DYRK1A knockdown studies” Figure 4 legend and new Figure 4.

In addition, both reviewers would like you to show that the compounds have similar effects on more than 1 huES line.

We have now specified more clearly what hESC lines were used in which studies. Three lines were used in the study: HES-3 *PAX6^mCherry^*as the reporter cell line for induction assays, Genea022 for the CRISPR DYRK1A activation studies, and WA09 for most additional work. Although we did not carry out a specific experiment with all three lines, there is sufficient overlap in the endpoints to demonstrate that the effect is not cell lines specific. We have identified the cell lines used in particular experiments in the Figure legends of Figure 2, Figure 3 and 4.

Reviewer 2 has requested a confirmatory study (point 2) that seems important to carry out since it would provide strong support for the mechanisms that you propose.

Please see the response to the reviewer below.

Finally, we have added citations to three new clinical genetics studies (5-7) published since our original report that provide further strong evidence that DYRK1A haploinsufficiency results in severe intrauterine growth retardation and microcephaly in the human (Introduction, second paragraph). These studies further highlight the relevance of our work to human neurodevelopmental genetics.

Reviewer #1:[…] 1) The methods for the kinase assays in Table 1 and 2 should be detailed, including the source of kinase protein preparation, substrate peptides, K_m_ for each peptide substrate, and ATP? Is each assay performed at K_m_ for substrate and Km for ATP? As the IC_50_s of the chemical compounds depend on the kinase assay condition, such information must be provided, and critical in evaluating the results for specificity. Why the discrepancy between Table 1, and Table 2 in terms of the IC_50_s of ID8 against DYRK1A (0.054 vs. 0.104μM), GSK3b (0.45μM vs. 0.15μM)?

We have now included this information: see response to Editor point 2 above. The results in Tables 1 and 2 report independent experiments run separately. These experiments were performed by a commercial research organisation on two occasions approximately one year apart. The differences in absolute IC_50_ values reflect assay variation relating to preparations of different batches of enzyme and substrate, but the variation does not affect interpretations regarding specificity or relative potency.

2) For the evaluation of cellular activities of ID8 series compounds, dose responses of key compounds in this assay, should be included in the Table 2. Cellular toxicity needs to be addressed. At 5 μm, does ID-8 cause any reduction in the number of cultured cells (Figure 2, right panel)?

In our previous study (Hasegawa et al., (8)) we observed no toxicity of ID-8 up to 10 µM, and our experience in this study was similar.

CRISPR-mediated conditional deletion of DYRK1A would be a direct and conclusive genetic evidence to demonstrate that DYRK1A is necessary to support the neural specification, and should be included in the argument.

A key point. Please see response 4 to the Editors above.

3) It is not conclusive whether GSK3b underlies the effect of ID-8 on neural specification, as CHIR99021 has an effect on PAX reporter assay. CHIR99021 has a different profile across kinome, and the noticed difference on Oct-4 expression may be derived from off-target effects of CHIR99021. Providing a GSK3b dependent cellular readout with ID-8 compounds in these cells, such as p-tau, or p-CRMP, could help answer wither ID-8 compounds hit GSK3b in these cells.

Another critical issue. Please see the response to the Editors above point 1.

4) The author showed that ID-8 compounds blocked the neural specification, most likely through inhibiting DYRK1A. Previously, it is shown that ID-8 promotes the stem-ness of huES cells by promoting self-renewal (Hasegawa et al., 2012). Are the observations in this report merely reflecting the fact that the huES cells in the presence of ID8 have more stem-ness, hence resistance to be induced for any lineage specialization, or that DYRK1A is a key switch for neural lineage specification?

This is a key point. In Hasegawa et al. (8) we showed ID-8 did indeed promote self-renewal but this effect quite modest except when the compound was used in combination with WNT3A (please see Figure 1 in that manuscript). In fact, the most striking effect of ID-8 observed in that study was the inhibition of WNT3A induction of mesendodermal genes (Figure 1 in Hasegawa et al. (8)), though in retrospect we did not emphasize this enough in that paper. ID-8 in fact has minor effects under conditions that promote hESC self-renewal (in this study see Figure 2—figure supplement 1) but, as in Hasegawa et al. (8) the most prominent effect of the compound is to insulate of subset of hESC from powerful differentiation stimuli. We have added a brief note in the Discussion to emphasize that the effect of ID-8 in our previous study was modest except in the presence of WNT3A (Discussion, fifth paragraph).

Can huES cells be induced into neurons, in the presence of ID8, by overexpressing neurogenin2 that bypass the neuroprogenitor stage (Sudhof lab, 2013 Neuron)? In either case, the molecular targets need to be identified/discussed to provide a mechanism for such observation, are EGFR, FGF, p53 etc. involved in such a functional role (PMID 20237271; PMID 18771760; Ferron et al., 2010).

Though this question of direct reprogramming of ID-8 treated cells is interesting, the interaction of reprogramming factors with pathways altered by the drug is liable to be complex and not easily interpretable.

Reviewer #2:[…] 1) Inter-hESC line variability – in the Materials and methods, the authors describe using each of a HES3-knockin-PAX6 reporter, H9 and WA9 hESC lines. It would be helpful if they clarified which lines were used in which figures. Moreover, the manuscript would be improved if the effect of ID-8 was directly compared across all three hESC lines in a single figure panel (i.e. Pax6 qPCR or AB-based FACS).

We have clarified which cell lines were used in particular experiments, please see response 5 to Editors above.

2) NPCs: The authors suggest that ID-8 treatment is preventing hESCs from specifying to an NPC fate, supported by data showing ID8 doesn't change hESC replication or pluripotency markers (Figure 2—figure supplement 1). To confirm that this is a patterning defect, rather than a replication, death or differentiation phenotype, would it be possible to FACS purify remaining Pax6-GFP positive cells following ID8 treatment and assess their capacity to replicate and undergo neuronal differentiation (either with or without ID8 present)? Specifically, I am curious whether the remaining NPCs generate both neurons and astrocytes, and if neurons, both excitatory and inhibitory.

We did not look at this specific question in this study because we have previously shown that cells maintained in ID8 can be grown long term without toxicity or loss of pluripotency (8). It would be challenging to recover the PAX6 cells following flow cytometry without loss of viability. A key point of the present study is that DYRK1A insulates cells from potent signals that drive lineage specification, and our work here and in our previous study shows that this is achieved without toxicity, growth inhibition, or irreversible loss of pluripotency.

3) Disease relevance: The authors make repeated note of the presumed role of DYRK1A in neurodevelopmental disease, both in their Introduction and the Discussion, but fail to discuss any similarities between DYRK1A+\- hiPSC phenotypes and ID8 effects. Can they phenocopy DYRK1A+\- phenotypes across an ID8 dose-response curve? Even more important, can they rescue developmental effects in DS hiPSCs across an ID8 dose-response curve? (I understand that this would be the most difficult experiment of all the ones I have suggested, and that DS hiPSCs may not be available to this group – it is just the most interesting question I had in reading this manuscript.)

We speculate in the Discussion that the effects of ID-8 on pluripotent stem cells might account for much of the DYRK1A haploinsufficiency syndrome-namely that failure to specify sufficient neural progenitor cells might result in poor endowment of the CNS and severe effects on brain development observed in patients. Though we do not assert this directly, the experiment in Figure 4 suggests that the drugs could indeed reverse the effects of DYRK1A overexpression (the DS condition). We add a note to this effect in the Discussion (third paragraph).

a) Given that the authors demonstrated an ability to prevent neural differentiation with ID8, even when up-regulating DYRK1A, can they similarly impair neural differentiation by knocking down DYRK1A with dCas9-KRAB? This would confirm the specificity of the ID8 activity.

We now provide data showing that this is infact the case, see response to the Editors, point 4.

b) Similarly, DYRK1A is a member of the dual-specificity tyrosine phosphorylation-regulated kinase family. It must have some known downstream targets that the authors can query by western blot, etc., to confirm the specificity of ID-8. If this was done in previous publications, it should be noted and cited.

This is a solid suggestion. One of the challenges of studying this particular protein kinase is that it interacts with a myriad of signaling pathways. Future work will address downstream targets in hPSC but this is beyond the scope of the current study.

4) Technology relevance: Does the addition of ID8 improve the efficacy/yield of non-neuronal (i.e. cardiac, muscle, endoderm) differentiation protocols? Are there any known activators of DYRK1A that could be used to improve neuronal differentiation protocols?

It is unlikely that ID-8 would enhance other differentiation protocols. As shown in Hasegawa et al. (8) the compound profoundly inhibits the induction of mesendodermal differentiation by WNT3A. The results in Figure 4 indicate that indeed DYRK1A activation could enhance neural differentiation, but we are unaware of specific compounds with this effect.

1) Chambers SM, Mica Y, Lee G, Studer L, Tomishima MJ. Dual-SMAD

Inhibition/WNT Activation-Based Methods to Induce Neural Crest and Derivatives from Human Pluripotent Stem Cells. Methods Mol Biol 2016;1307:329-43.

2) Denham M, Bye C, Leung J, Conley BJ, Thompson LH, Dottori M. Glycogen synthase kinase 3beta and activin/nodal inhibition in human embryonic stem cells induces a pre-neuroepithelial state that is required for specification to a floor plate cell lineage. Stem Cells 2012;30(11):2400-11.

3) Denham M, Hasegawa K, Menheniott T, Rollo B, Zhang D, Hough S, et al. Multipotent caudal neural progenitors derived from human pluripotent stem cells that give rise to lineages of the central and peripheral nervous system. Stem Cells 2015;33(6):1759-70.

4) Menendez L, Yatskievych TA, Antin PB, Dalton S. Wnt signaling and a Smad pathway blockade direct the differentiation of human pluripotent stem cells to multipotent neural crest cells. Proc Natl Acad Sci U S A 2011;108(48):19240-5.

5) Dang T, Duan WY, Yu B, Tong DL, Cheng C, Zhang YF, et al. Autismassociated Dyrk1a truncation mutants impair neuronal dendritic and spine growth and interfere with postnatal cortical development. Mol Psychiatry 2017.

6) Evers JM, Laskowski RA, Bertolli M, Clayton-Smith J, Deshpande C, Eason J, et al. Structural analysis of pathogenic mutations in the DYRK1A gene in patients with developmental disorders. Hum Mol Genet 2017;26(3):519-26.

7) van Bon BW, Coe BP, Bernier R, Green C, Gerdts J, Witherspoon K, et al. Disruptive de novo mutations of DYRK1A lead to a syndromic form of autism and ID. Mol Psychiatry 2016;21(1):126-32.

8) Hasegawa K, Yasuda SY, Teo JL, Nguyen C, McMillan M, Hsieh CL, et al. Wnt signaling orchestration with a small molecule DYRK inhibitor provides long-term xeno-free human pluripotent cell expansion. Stem Cells Transl Med 2012;1(1):1828.

9) van de Leemput J, Boles NC, Kiehl TR, Corneo B, Lederman P, Menon V, et al. CORTECON: a temporal transcriptome analysis of in vitro human cerebral cortex development from human embryonic stem cells. Neuron 2014;83(1):51-68.

[Editors' note: further revisions were requested prior to acceptance, as described below.]

After additional discussion with the reviewers, we reached the conclusion that you should still carry out the experiment highlighted as reviewer 2's point #1 (repeating your main assay using 2 additional lines). We believe that this experiment is straightforward to do and will be important in convincing readers of the robustness of your observations.Reviewer #2:In their revised manuscript, Bellmaine et al. now better demonstrate that inhibition of DYRK1A prevents hESCs from differentiating to NPCs. The authors were reasonably responsive to my concerns, particularly in adding a genetic inhibition of DYRK1A to complement the pharmacological studies. Nonetheless, two simple experiments remain.

We now include new data showing that during dual SMAD induction of neural specification, DYRK1A inhibition by four of our active inhibitors reduces the level of *PAX6* mRNA transcript induction in three of the four cell lines used in this study (WA09, GENEA022, and C11, an iPSC line used in new confirmatory experiments (below). The Wolvetang laboratory, which carried out this additional work, does not currently have access to the fourth cell line used in the study (HES-3 *PAX6^mCherry^*). These data are found in new Figure 4—figure supplement 1, reported in the Results and Material and methods sections The C11 cell line is referenced (Briggs et al., 2013). We also present new DYRK1A shRNA knockdown data on additional cell lines, see response to Reviewer 2 comment 1 below.

1) Inter-hESC line variability – the authors did a great job clarifying which hESC lines were used in each figure panel. Nonetheless, I still think it would be helpful to compare all three hESC lines across at least a single assay (i.e. Pax6 qPCR or FACS), in order to assess how robust the effects are across hESC lines.

Please see Essential revisions above. We now also present *DYRK1A* shRNA knockdown data on two additional cell lines, Genea22 (previously used for the DYRK1A induction study) and another iPSC line, C11. These data confirm the previous shRNA data obtained with WA09. These data are included in new Figure 4—figure supplement 1., reported in the last paragraph of the Results subsection “ID-8 offsets effects of DYRK1A overexpression on neural specification of hESC and DYRK1A knockdown phenocopies effects of inhibitors”, the shRNA methods are the same as those described for WA09, and the C11 cell line is referenced Briggs et al., 2013.

2) NPCs: The authors suggest that ID-8 treatment is preventing hESCs from specifying to an NPC fate. It would be interesting to collect the remaining Pax6-GFP positive cells following ID8 treatment (or shRNA knockdown) and assess either their gene expression and/or their capacity to replicate and undergo neuronal differentiation (either with or without ID8 present)? Specifically, I am curious whether the remaining NPCs generate both neurons and astrocytes, and if neurons, both excitatory and inhibitory.

We accept that this would be an interesting experiment to carry out, however, the isolation of these cells presents technical challenges as noted previously, and is beyond the scope of this study which specifically addresses the inhibition of neural specification.